# Nucleotide sugar biosynthesis occurs in the glycosomes of procyclic and bloodstream form *Trypanosoma brucei*

**Maria Lucia Sampaio Guther[1], Alan R. Prescott[2], Sabine Kuettel[1], Michele Tinti[1], Michael A. J. Ferguson[1]** *

1 Wellcome Centre for Anti-Infectives Research, School of Life Sciences, University of Dundee, Dundee, United Kingdom, 2 Dundee Imaging Facility, School of Life Sciences, University of Dundee, Dundee, United Kingdom

* m.a.j.ferguson@dundee.ac.uk

## Abstract

In *Trypanosoma brucei*, there are fourteen enzymatic biotransformations that collectively convert glucose into five essential nucleotide sugars: UDP-Glc, UDP-Gal, UDP-GlcNAc, GDP-Man and GDP-Fuc. These biotransformations are catalyzed by thirteen discrete enzymes, five of which possess putative peroxisome targeting sequences. Published experimental analyses using immunofluorescence microscopy and/or digitonin latency and/or subcellular fractionation and/or organelle proteomics have localized eight and six of these enzymes to the glycosomes of bloodstream form and procyclic form *T. brucei*, respectively. Here we increase these glycosome localizations to eleven in both lifecycle stages while noting that one, phospho-N-acetylglucosamine mutase, also localizes to the cytoplasm. In the course of these studies, the heterogeneity of glycosome contents was also noted. These data suggest that, unlike other eukaryotes, all of nucleotide sugar biosynthesis in *T. brucei* is compartmentalized to the glycosomes in both lifecycle stages. The implications are discussed.

## Author summary

All eukaryotes add sugar chains to proteins to make glycoproteins, most of which decorate the cell surface and play central roles in how cells interact with their environment and with other cells. These sugar chains are built up using nucleotide sugars to donate the individual sugars. The nucleotide sugars themselves are generally made in the cytoplasm of cells but in *Trypanosoma brucei*, the causative agent of human African trypanosomiasis and nagana in cattle, they are made inside small organelles called glycosomes. This very unusual arrangement in parasite metabolism is notable and may offer therapeutic opportunities.

## Introduction

The tsetse-fly transmitted protozoan parasite *Trypanosoma brucei* is responsible for human and animal African trypanosomiasis. The bloodstream form (bsf) of this organism depends on

**Data Availability Statement:** Immunofluorescence microscopy localisation of nucleotide sugar biosynthetic enzymes in procyclic form T. brucei (wide-field images) are available at DOI: 10.6084/

m9.figshare.13124909. The mass spectrometry proteomics data have been deposited to the ProteomeXchange Consortium via the PRIDE partner repository with the dataset identifier PXD023124.

**Funding:** This work was funded by The Wellcome Trust through an Investigator Award (10842/Z/13/Z) to MAJF. The funders had no role in study design, data collection and analysis, decision to publish, or preparation of the manuscript.

**Competing interests:** The authors have declared that no competing interests exist.

a surface coat made of glycosylphosphatidylinositol (GPI) anchored and *N*-glycosylated variant surface glycoprotein (VSG) to evade the host innate immune system and the acquired immune system through antigenic variation [1]. The bsf parasite also expresses many lower abundance glycoproteins, such as a novel transferrin receptor (TfR) [2–4], a lysosomal/endosomal protein called p67 [5], the invariant surface (ISG) and endoplasmic reticulum (IGP) glycoproteins [6,7], the Golgi/lysosomal glycoprotein tGLP-1 [8], the membrane-bound histidine acid phosphatase TbMBAP1 [9], the flagellar adhesion zone glycoproteins Fla1-3 [10,11], the flagellar pocket/endosomal system haptoglobin-hemoglobin receptor (HpHbr) [12] and serum resistance antigen (SRA) [13], the complement factor H receptor (FHR) [14] and others, such as the metacyclic trypomastigote-specific ISG [15]. Some of these are metacyclic and/or bsf specific glycoproteins (eg. VSG, TfR, ISG, TbMAP1, HpHbr, SRA, FHR) while others are also common to the tsetse midgut-dwelling procyclic form (pcf) of the parasite. Further, pcf parasites also express unique glycoproteins, notably the abundant GPI-anchored procyclins, some of which are *N*-glycosylated [16,17], and a high-molecular weight glycoconjugate [18,19]. Many of the *N*-glycan structures expressed by bsf *T. brucei* have been solved and these include conventional oligomannose and biantennary complex structures as well as paucimannose and extremely unusual 'giant' poly-*N*-acetyl-lactosamine (poly-LacNAc) containing complex structures [20–23]. In contrast, only oligomannose *N*-glycans have been structurally described in wild type pcf trypanosomes [16,24]. The GPI anchor structures of several bsf VSGs [25–28] and of the TfR [5,29] have also been solved, as have those of pcf procyclins [16]. Both bsf and pcf GPI anchors contain the canonical conserved GPI core structure but they are the most divergent among the eukaryotes in terms of their carbohydrate sidechains, containing up to 1 βGal and 5 αGal residues in the bsf GPI sidechains and multiple, branched, *N*-acetyllactosamine and lacto-*N*-biose repeats capped with α2–3 sialic acid in the pcf GPI sidechains [16,27,30,31]. Recently, the *O*-glycosylation (via novel Glcα1-*O*-Ser linkages) of certain VSG variants has also been described [32]. The entire repertoire of known *T. brucei N*-linked, *O*-linked and GPI glycans is composed exclusively of the monosaccharides mannose, galactose, glucose, N-acetylglucosamine, glucosamine and sialic acid. In addition, a trace of fucose has been reported in the pcf high-molecular weight glycoconjugate [19], and recent data suggest that fucosylation may occur inside the parasite mitochondrion [33]. These combined monosaccharide compositions are consistent with the nucleotide sugar biosynthetic capacity of bsf and pcf trypanosomes [34] (Fig 1), apart from the absence of CMP-sialic acid. However, this apparent discrepancy is because sialic acid is added to the procyclin GPI sidechains via cell-surface transialidase enzymes that transfer α2–3 sialic acid from host glycoconjugates in a nucleotide sugar-independent reaction [35,36]. The non-*N*-acetylated glucosamine in the GPI anchors originates from UDP-GlcNAc and is the product of the GPI pathway GlcNAc-PI de-*N*-acetylase [37,38]. Although mature *T. brucei N*-glycans do not contain glucose, the trypanosome *N*-glycans undergo reversible glucosylation from UDP-Glc in the endoplasmic reticulum (ER) as part of a UDP-Glc: unfolded glycoprotein glucosyltransferase (UGGT) / α-glucosidase II / calreticulin ER quality control system [23,39]. In *T. brucei* UDP-Glc is also the donor for the biosynthesis of base J [40], the obligate precursor of UDP-Gal (via UDP-Glc 4'-epimerase (TbGALE) [41]), and presumed to be the donor for VSG *O*-glycosylation [32].

Nucleotide sugar biosynthesis in *T. brucei* is quite conventional, in so far as homologues of most of the necessary enzymes can be found by BLASTp searches with corresponding prokaryotic and/or eukaryotic amino acid sequences [34]. The only exception to this is the absence of a canonical phosphoglucose mutase (PGM) enzyme, the function of which is redundantly replaced by the parasite phosphomannose mutase (TbPMM) and phospho-N-acetylglucosamine mutase (TbPAGM) enzymes [42]. Further, the following trypanosome enzymes of nucleotide sugar biosynthesis have been expressed in *E. coli* and shown, where determined, to have

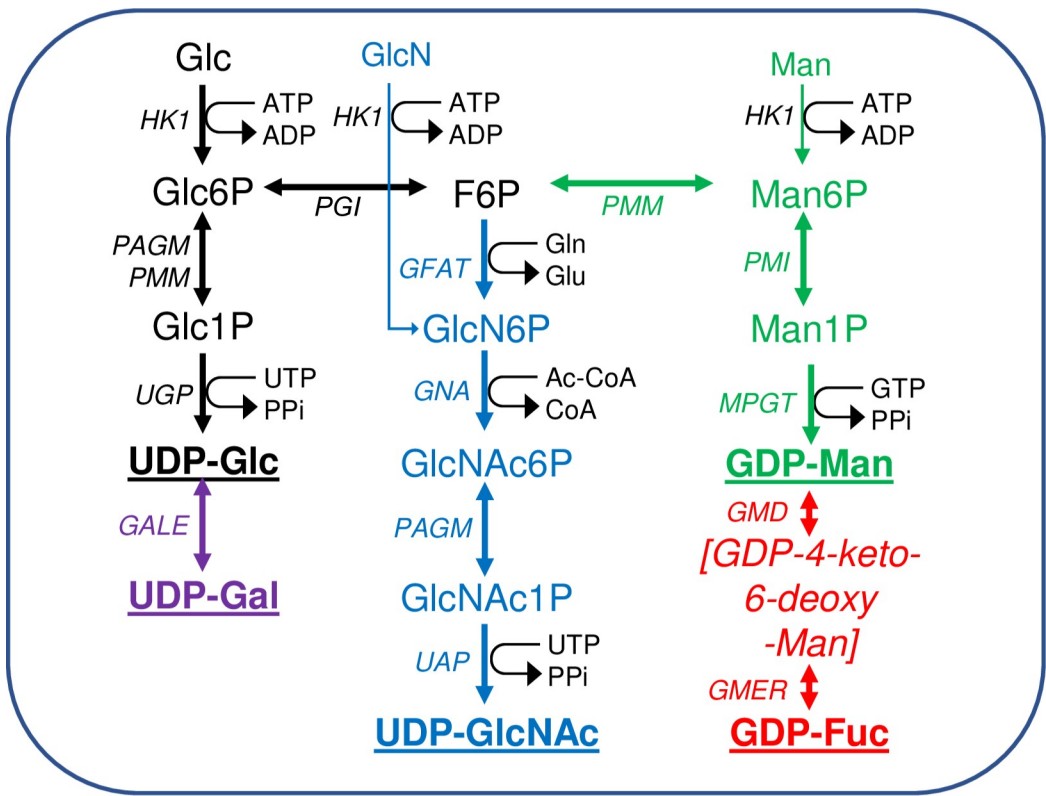

**Fig 1. Scheme of nucleotide sugar biosynthesis in *T. brucei*.** Nucleotide sugar biosynthesis in *T. brucei* according to [34], showing the 14 biotransformations and 13 enzymes involved. No dedicated phospho-glucose mutase (PGM) gene exists in T. brucei and the interconversion of Glc6P and Glc1P is performed by PAGM and/or PMM [42]. The enzyme abbreviations (*in italics*) appear in the Introduction. The terminal nucleotide sugars used for glycosylation reactions are in bold and underlined. Information on enzyme essentiality, protein and antibody production, and sub-cellular localisation appear in (Table 1). The main source of all of the nucleotide sugars is Glc but both Man and GlcN can enter their respective pathways via hexokinase (HK1).

typical kinetic properties and conserved three dimensional structures (Table 1): UDP-glucose 4'-epimerase (TbGALE) [43,44]. UDP-glucose pyrophosphorylase (TbUGP) [45]. Phospho-mannose isomerase (TbPMI) [46]. Phosphomannose mutase (TbPMM) [42,47]. Mannose phosphate guanyltransferase (TbMPGT) [47,48]. Glucosamine-6-phosphate N-acetyltransfer-ase (TbGNA) [49]. Phospho-N-acetylglucosamine mutase (TbPAGM) [42]. UDP-N-acetylglu-cosamine pyrophosphorylase (TbUAP) [50,51]. GDP-mannose dehydratase (TbGMD) [52]. GDP-4-dehydro-6-deoxy-D-mannose epimerase/reductase (TbGMER) [52]. However, where nucleotide sugar biosynthesis in *T. brucei* diverges most dramatically from the norm is with respect to its subcellular location. Thus, whereas eukaryotic nucleotide sugar biosynthesis is generally either known or assumed to occur in the cytoplasm, for *T. brucei* many of these reactions appear to occur in the glycosomes [42,44–46,49,50,52–54] (Table 1). Glycosomes are kinetoplastid peroxisome-like membrane-bound organelles, named such because they contain enzymes of glycolysis [55–59,66–68], including hexokinase (TbHK1) and phospho-glucose isomerase (TbPGI) that are also components of the *de novo* pathways to the nucleotide sugars.

Here, we analyse the subcellular locations of nine nucleotide sugar biosynthetic enzymes in pcf trypanosomes by immunofluorescence microscopy using mono-specific mouse polyclonal antibodies raised to purified recombinant, soluble and enzymatically active proteins. Further, we use eight of these nine antibodies to determine the digitonin latency of the corresponding

**Table 1. Summary of data on *T. brucei* nucleotide sugar biosynthetic enzymes.** Data are from the literature [#] or from this paper [TP]. GL = glycosomal; CP = cytoplasmic; IFM = immunofluorescence microscopy; *indicates that IFM was performed using an epitope tag rather than an antibody to the whole protein; PTS = peroxisome targeting sequence (type 1 or type 2) for the respective *T. brucei* (Tb), *T. cruzi* (Tc) and *L. major* (Lm) enzymes [50,65]. Enzyme abbreviations are listed in the Introduction, with the exception of TbGFAT, glutamine: fructose-6-phosphate aminotransferase.

| Enzyme | Gene | Essential for cell survival | | Protein expression, crystallography and antibodies | | | IFM | | Digitonin Latency | | Sub-cellular Fractionation | | Glycosome proteome | | PTS | | |
|---|---|---|---|---|---|---|---|---|---|---|---|---|---|---|---|---|---|
| | | bsf | pcf | Prot | Xtal | Ab | bsf | pcf | bsf | pcf | bsf | pcf | bsf | pcf | Tb | Tc | Lm |
| TbHK1 | Tb927.10.2010 | [62] [63] | | [60] | | [59] | GL [59] | GL [59] | GL [58] [59] | GL [57] [59] | GL [55] [56] | GL [57] [58] | GL [53] | GL [54] [TP] | 2 | 2 | 2 |
| TbPGI | Tb927.1.3830 | | | [64] | [64] | | | | GL [59] | | GL [55] [56] | GL [57] | | GL [53] [54] [TP] | 1 | 1 | 1 |
| TbUGP | Tb927.10.13130 | [45] | | [45] | [45] | [45] | GL [45] | GL [TP] | GL [TP] | GL [TP] | | GL [TP] | | GL [54] | | | 1 |
| TbGALE | Tb927.11.2730 | [41] | [44] | [41] | [43] | [44] [TP] | GL [44] | GL [44] [TP] | GL [TP] | GL [TP] | GL [44] | | | GL [54] | 1 | | 2 |
| TbGFAT | Tb927.7.5560 | | | | | | | | | | | | | | | | |
| TbGNA | Tb927.11.11100 | [49] | | [49] | [49] | [49] | GL [49] | GL [TP] | | | | | | | | | |
| TbPAGM | Tb927.8.980 | [42] | | [42] | | [42] | GL [42] | GL+CP [TP] | GL+ CP [42] | GL+ CP [TP] | | | | | | | |
| TbUAP | Tb927.11.2520 | [50] | | [50] | [50] | [50] [TP] | GL [50] | GL [TP] | GL [TP] | GL [TP] | | | | GL [54] | 1 | 1 | 1 |
| TbPMI | Tb927.11.14780 | [46] | | [46] | | [46] | GL [46] | GL [TP] | GL [TP] | GL [TP] | | GL [TP] | | GL [53] [54] [TP] | 1 | 1 | |
| TbPMM | Tb927.10.6440 | [42] | | [42] | [42] | [42] | GL [42] | GL [TP] | GL [42] [TP] | GL [TP] | | | | | | 1 | 1 |
| TbMPGT | Tb927.8.2050 | [47] | | [47] [48] [TP] | | [TP] | CP [48] | GL [TP] | GL [TP] | GL [TP] | | GL [TP] | | | | | |
| TbGMD | Tb927.10.15490 | [52] | [52] | [52] | | | | GL [52]* | | | | | | | | | |
| TbGMER | Tb927.11.13990 | | | [52] | | [TP] | | GL [TP] | GL [TP] | GL [TP] | | | | | | | |

enzymes in both pcf and bsf trypanosomes. In addition, we localise three representative NS biosynthetic enzymes (TbPMI, TbMPGT and TbUGP) to the glycosome-containing small granular fraction by subcellular fractionation, and demonstrate heterogeneity in glycosome composition by density gradient centrifugation and proteomics.

## Results

### Subcellular localization by immunofluorescence microscopy

In previous work, we have purified the following soluble, enzymatically active, recombinant nucleotide sugar biosynthetic enzymes: TbUGP, TbGALE, TbGNA, TbPAGM, TbUAP, TbPMI and TbPMM (Table 1). With these, we have prepared either rabbit (TbGALE) or mouse polyclonal antibodies for immunofluorescence microscopy (IFM) localization studies

**Table 2. Antibodies to nucleotide sugar biosynthetic enzymes used in this study.** [TP] = this paper.

| Enzyme | Antibody species | Affinity purified | Mono-specific by Western | Reference |
|---|---|---|---|---|
| TbUGP | Mouse | Yes | Yes | [45] |
| TbGALE | Mouse | No | Yes | [TP] |
| TbGNA | Mouse | Yes | Yes | [TP] |
| TbPAGM | Mouse | No | Yes | [42] |
| TbUAP | Mouse | Yes | Yes | [TP] |
| TbPMI | Mouse | No | Yes | [46] |
| TbPMM | Mouse | No | Yes | [42] |
| TbMPGT | Mouse | No | Yes | [TP} |
| TbGMER | Mouse | No | Yes | [TP] |

against bsf trypanosomes (Table 1). To these we have added new polyclonal mono-specific mouse antibodies to recombinant TbGALE, TbGNA, TbUAP, TbMPGT and TbGMER (see Methods). In all cases, the mouse antibodies were affinity-purified against immobilized immunogen and/or demonstrated to be mono-specific by Western blotting (Table 2; S2 Fig). These antibodies were used in immunofluorescence microscopy against fixed and permeabilized pcf trypanosomes, along with rabbit antibodies to definitive glycosomal (TbGAPDH) and cytosolic (TbEnolase) markers (Figs 2, 3, and 4). Wider field images containing several cells are available in (10.6084/m9.figshare.13124909). In all cases, except for TbPAGM (Fig 2, middle panels), the mono-specific mouse antibodies produced punctate staining that was coincident, at least in part, with the rabbit anti-TbGAPDH antibody staining, suggesting glycosomal locations. In contrast, there was no overlap with cytoplasmic anti-TbEnolase immunostaining. In the case of TbPAGM, the immunostaining suggested both glycosomal and cytosolic localization. Interestingly, the punctate co-staining with anti-NS biosynthetic enzyme and anti-TbGAPDH antibodies was imperfect, in so far as some puncta were more red or green than yellow. This implies heterogeneity in glycosomal contents, which is discussed later. However, although the ratios of TbGAPDH (red) to NS enzyme (green) signals vary widely, quantitative analysis (Table 3) shows that the majority of the NS enzyme signals colocalize with punctate TbGAPDH signals, with the exception of TbPAGM.

## Subcellular localization by digitonin latency

Digitonin latency is a powerful adjunct technique to assess whether soluble proteins are either cytosolic or sequestered in membrane-bound intracellular organelles. Digitonin is a steroidal-saponin detergent-like natural product that first permeabilizes and subsequently solubilizes biological membranes. Digitonin has a preference for high sterol content membranes. Thus, cytosolic soluble proteins are liberated from cells at very low digitonin concentrations, as the sterol-rich plasma membrane is the first affected, and organellar contents are subsequently liberated at significantly higher digitonin concentrations. The requirement for a higher digitonin concentration to effect protein release is known as 'digitonin latency'. This method has proved extremely useful in discriminating cytosolic versus glycosomal proteins in trypanosomatids, for example [42,58–60,61,69]. In our studies, we used the aforementioned mono-specific polyclonal mouse antibodies to eight nucleotide biosynthetic enzymes, along with rabbit antibodies to authentic glycosome lumen (TbGAPDH or TbAldolase), glycosome membrane (TbPEX13.1) and soluble cytosolic (TbEnolase) markers to assess digitonin latency of these enzymes in pcf and bsf trypanosomes (Fig 5A and 5B, respectively). As expected, the cytosolic marker, TbEnolase, was released from both lifecycle stages at very low digitonin

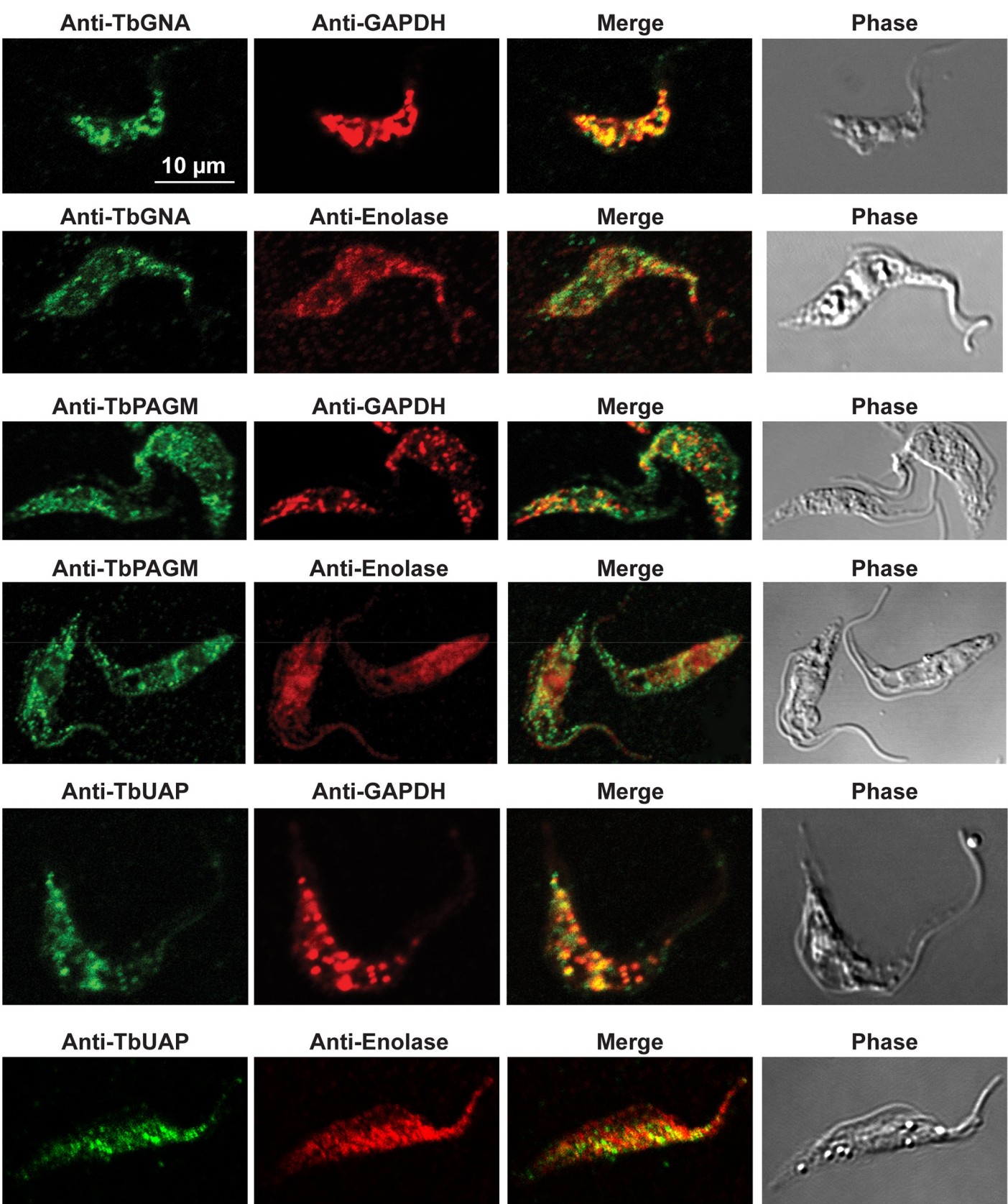

**Fig 2. Immunofluorescence microscopy localisation of UDP-GlcNAc biosynthetic enzymes in procyclic form *T. brucei*.** Enzymes of UDP-GlcNAc biosynthesis were localised with mono-specific mouse antibodies (green channels). Authentic glycosome and cytosol markers TbGAPDH and TbEnolase, respectively, were localised with specific rabbit antisera (red channels). Merged green and red channels and phase-contrast images are also shown.

concentrations whereas TbAldolase and TbGAPDH were not quantitatively released until 0.1% TX100 was used. This was not because the glycosome membrane was not being gradually solubilized by the digitonin, since the gradual release of membrane protein TbPEX13 was evident in these experiments, but because of the possibly aggregated nature of these very abundant proteins in what has been proposed to be a dense crystalloid core inside of glycosomes [56]. In contrast, TbPAGM showed a biphasic release pattern, consistent with dual cytoplasmic and glycosomal location. All the other nucleotide sugar biosynthetic enzymes showed release starting around 0.04 mg digitonin / mg protein, consistent with them being predominantly or exclusively glycosomal proteins. Interestingly, in all cases, the proteins seem to be slightly harder to release with digitonin from pcf cells than from bsf cells.

## Subcellular localization by subcellular fractionation and evidence for glycosome heterogeneity by density gradient centrifugation and proteomics

Procyclic cells were lysed using a high-pressure homogenizer [54] and the lysate was subjected to differential centrifugation according to [70]. This procedure results in a nuclear fraction, a large granular fraction (LG), a small granular fraction (SG) and a cytoplasmic fraction (C), with the latter also containing microsomes of the plasma membrane, Golgi apparatus and endoplasmic reticulum (ER). These fractions were subjected to SDS-PAGE and Western blotting and the blots were probed with antibodies to authentic glycosomal, mitochondrial, cytoplasmic and ER markers (TbGAPDH, TbGAP1 [71], TbENO and calreticulin, respectively) which confirmed that the majority of mitochondrial TbGAP1 was in the LG fraction, that the majority of cytoplasmic TbENO and ER calreticulin was in the C fraction and that the majority of glycosomal TbGAPDH was in the SG fraction, as expected [70] (Fig 6A). Antibodies to three representative NS biosynthetic enzymes (TbPMI, TbMPGT and TbUGP) showed that these enzymes are found exclusively to the glycosome-containing SG fraction (Fig 6A, Table 1), consistent with the immunofluorescence microscopy and digitonin latency data described above.

The same SG fraction was further subjected to iodixanol density gradient centrifugation. Two opalescent bands (fractions 1 and 2) were observed (Fig 6B) at densities of 1.15 and 1.18 g/mL, similar to that reported for rat liver peroxisomes using iodixanol (1.175 g/mL) [72]. Each was isolated and, following reduction, alkylation and trypsin digestion of the protein content, subjected to proteomic analysis by liquid chromatography-tandem mass spectrometry (LC-MS/MS). The protein groups identified in both fractions were highly-enriched for glycosomal proteins, as expected. Intensity-based absolute quantitation (iBAQ) values were extracted for the protein groups belonging to a reference high-confidence glycosomal proteome [54]. These protein groups (x axis) are plotted against three iBAQ scales, from low abundance (iBAQ values $<1 \times 10^4$) to high-abundance (iBAQ values $>3 \times 10^6$), in (Fig 6C). Consistent with its higher density, fraction 2 contains more glycosomal protein than fraction 1. However, the relative abundances of glycsomal proteins between the two fractions are different, suggesting compositional heterogeneity between them.

## Discussion

Evidence that NS biosynthetic enzymes might be glycosomal first came from the sequencing of the gene encoding TbGALE, the enzyme interconverts UDP-Glc and UDP-Gal. The predicted

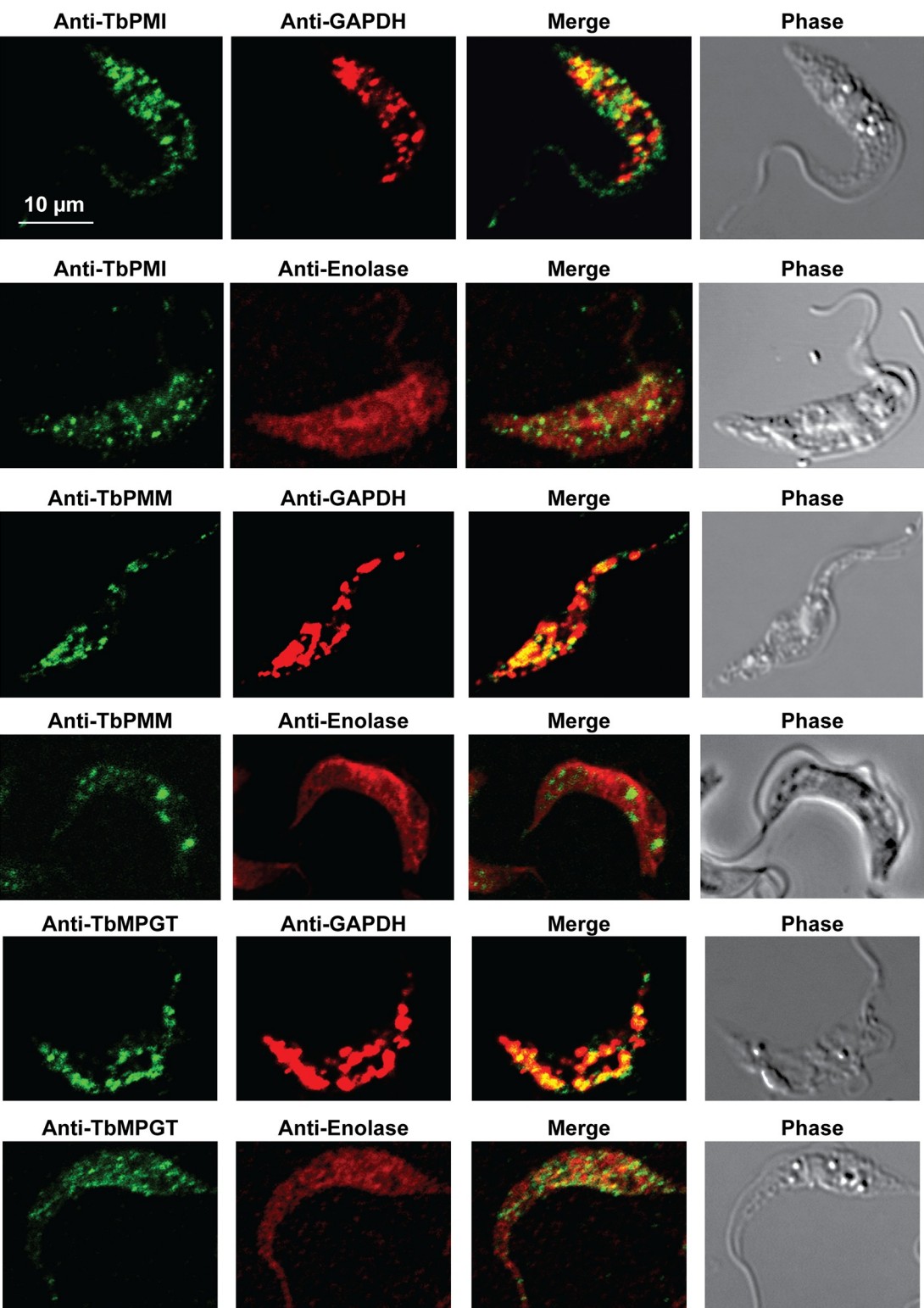

**Fig 3. Immunofluorescence microscopy localisation of GDP-Man biosynthetic enzymes in procyclic form *T. brucei*.** Enzymes of GDP-Man biosynthesis were localised with mono-specific mouse antibodies (green channels). Authentic glycosome and cytosol markers TbGAPDH and TbEnolase, respectively, were localised with specific rabbit antisera (red channels). Merged green and red channels and phase-contrast images are also shown.

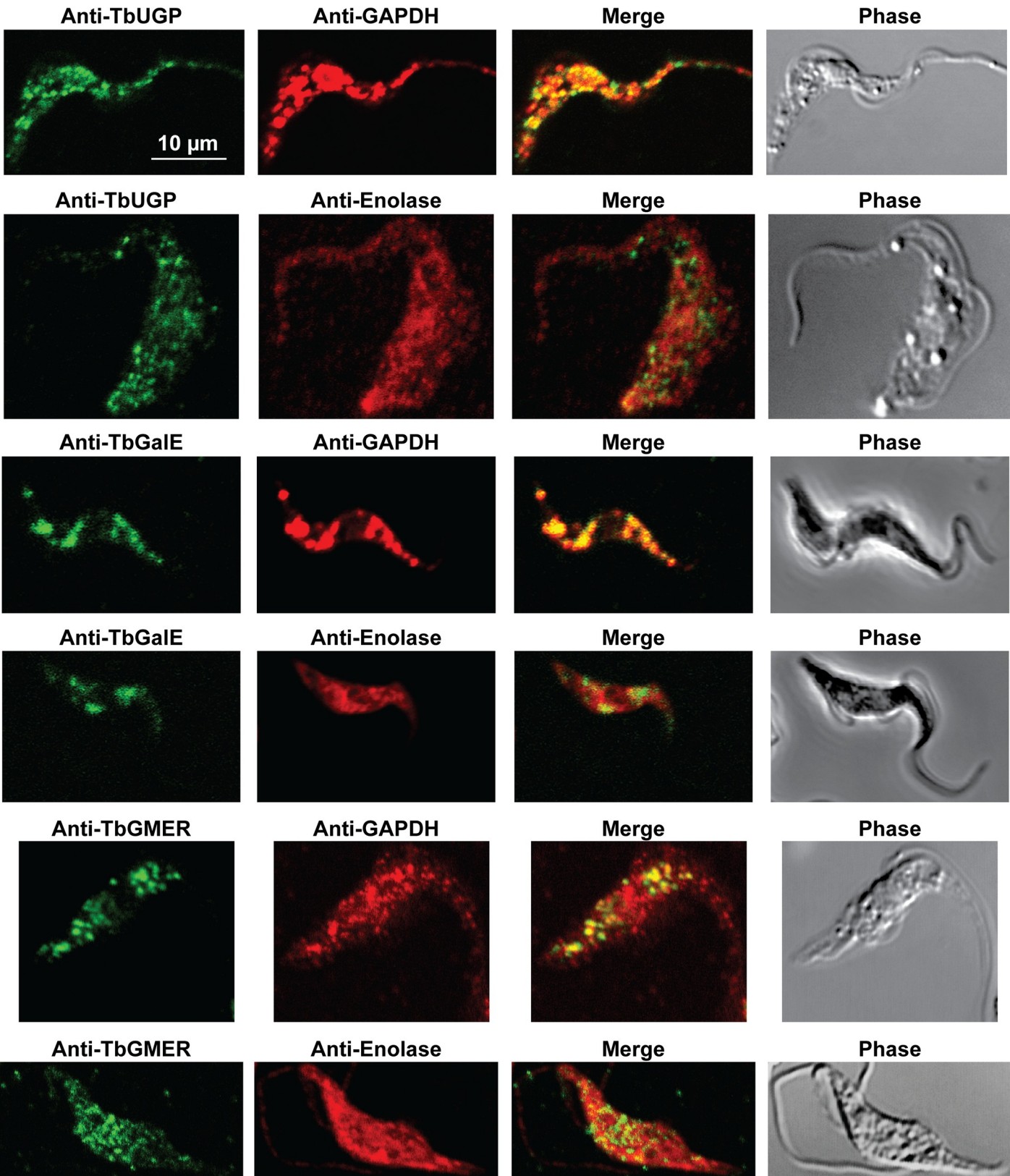

**Fig 4. Immunofluorescence microscopy localisation of UDP-Glc, UDP-Gal and GDP-Fuc biosynthetic enzymes in procyclic form *T. brucei*.** Enzymes of UDP-Glc, UDP-Gal and GDP-Fuc biosynthesis were localised with mono-specific mouse antibodies (green channels). Authentic glycosome and cytosol markers

TbGAPDH and TbEnolase, respectively, were localised with specific rabbit antisera (red channels). Merged green and red channels and phase-contrast images are also shown.

amino acid sequence contained a C-terminal peroxisomal targeting sequence type 1 (PTS1) of -TKL [41]. The glycosomal location of TbGALE was later confirmed experimentally in bsf and pcf cells [44]. A comprehensive screen of kinetoplastid genes for predicted PTS1 and N-terminal PTS2 sequences [65] added TbHK1 (PTS2), TbPGI (PTS1) and TbPMI (PTS1). TbUAP was later found to possess an atypical PTS1 sequence (-SNM) [50]. Interestingly, the HK1, GPI, UAP, and PMM enzymes of *T. cruzi* and *Leishmania major* also possess PTS motifs, as do the PMI and UGP enzymes of *T. cruzi* and *L. major*, respectively (Table 1). While many glycosomal components possess PTS signals for glycosomal import, many others do not. Internal targeting sequences have been proposed for *T. cruzi* phosphoglucomutase and *T. brucei* triosephosphate isomerase [73,74] but these are not amenable to bioinformatic prediction. Other peroxisomal proteins are imported by a 'piggyback' mechanism through association with PTS-bearing proteins [75]. In this context, analysis of the protein complex database for pcf cells, using the cluster explorer function, suggests that TbPMM may piggyback on PTS1-containing TbPGI since both appear in cluster 314 in SEC300 gel-filtration and cluster 274 in SEC1000 gel-filtration [76]. The mapping of PTS1- and/or PTS2 -containing proteins to the high-confidence proteome for pcf glycosomes suggested a sensitivity and specificity of about 40% and 50%, respectively, for the bioinformatic prediction of glycosomal location [54]. Consequently, experimental confirmations of helpful PTS1/2 glycosomal location predictions are highly desirable.

Previously, the majority of immunofluorescence microscopy (IFM) localisation data for *T. brucei* NS biosynthetic enzymes were for bsf trypanosomes, with only TbGALE and TbGMD localised by IFM in pcf cells (Table 1). Now, all of the NS biosynthetic enzymes in pcf cells, except for TbPGI (that contains a PTS1 sequence and that is known to be glycosomal from sub-cellular fractionation) and for TbGFAT (which has proven difficult to express as a soluble protein for antibody production) have been successfully localised to the glycosome by IFM. The majority of these localisations, both for bsf and pcf cells, are further strongly supported here by digitonin latency experiments (Fig 5; Table 1). Further, subcellular fractionation and Western blotting for three representative NS biosynthetic enzymes (TbPMI, TbMPGT and

**Table 3. Quantitative analysis of anti-NS enzyme and anti-TbGAPDH colocalization by IFM in pcf cells.**

| NS Enzyme | n[a] | Mander's colocalization coefficient[b] |
|---|---|---|
| TbGNA | 6 | 0.85 ± 0.05 |
| TbPAGM | 5 | 0.33 ± 0.06 |
| TbUAP | 5 | 0.89 ± 0.02 |
| TbPMI | 9 | 0.96 ± 0.01 |
| TbPMM | 4 | 0.73 ± 0.06 |
| TbMPGT | 4 | 0.89 ± 0.04 |
| TbUGP | 8 | 0.89 ± 0.03 |
| TbGalE | 4 | 0.88 ± 0.05 |
| TbGMER | 7 | 0.88 ± 0.05 |

a n is number of cells uses in the analysis.

b The Mander's colocalization coefficient (± standard deviation of the mean) is a measure of the amount of green (anti-NS enzyme) signal above background that colocalizes with red (anti-TbGAPDH) stained compartments.

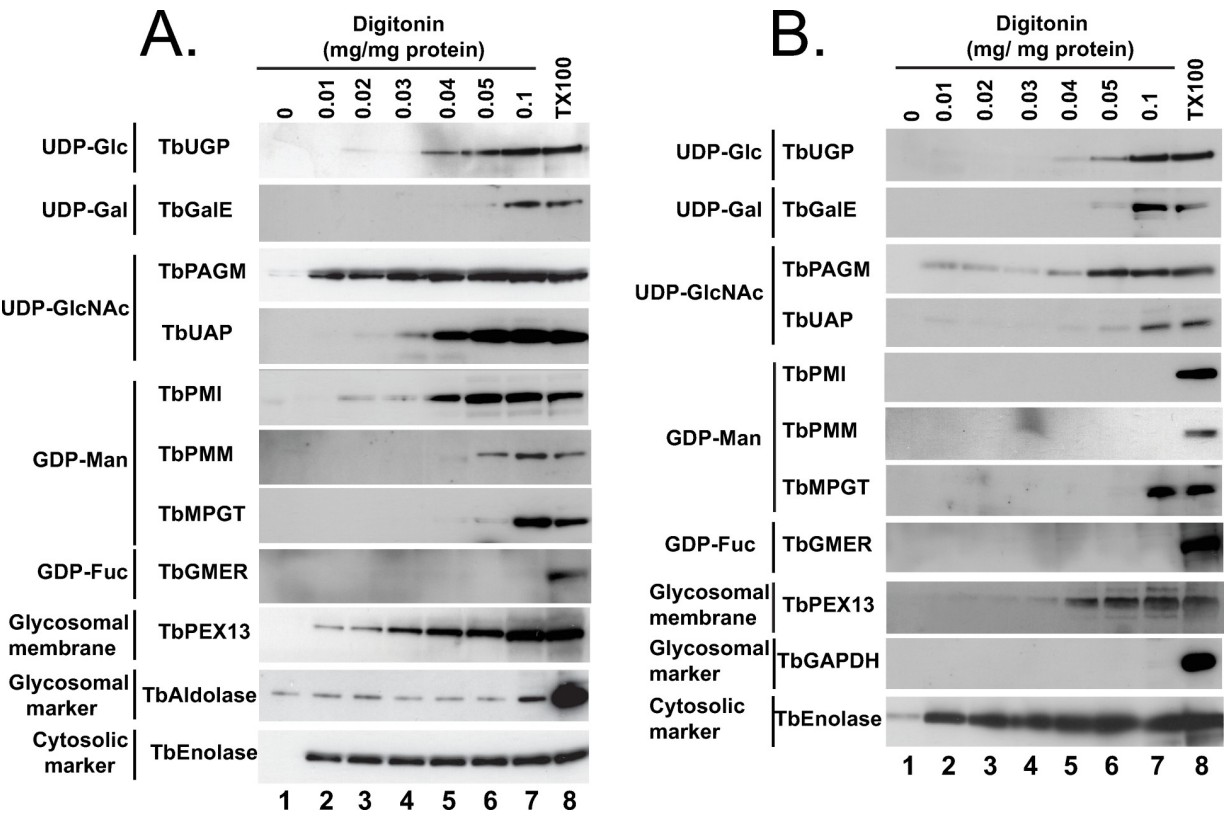

**Fig 5. Digitonin latencies of nucleotide sugar biosynthetic enzymes in bloodstream and procyclic forms of *T. brucei*.** Trypanosomes (bsf, panel A; pcf, panel B) were treated with increasing concentrations of digitonin (0.01 to 0.1 mg digitonin per mg trypanosome protein) and with no detergent and 0.1% TX100 as controls for no cell permeabilization and total cell and glycosome membrane permeabilization, respectively. The release of the cytoplasmic marker TbEnolase at the lowest digitonin concentration is consistent with the selective permeabilization of the plasma membrane over the glycosome membrane. The release of TbAldolase (panel A) or TbGAPDH (panel B) at higher digitonin concentrations, or only with 0.1% TX100, i.e., showing latency, is consistent with their known glycosome lumen locations. From the patterns of digitonin latency of the nucleotide sugar biosynthetic enzymes, we conclude that all shown, except TbPAGM, are principally or exclusively protected against release by digitonin by being sequestered in a low-sterol intracellular membrane-bound compartment.

TbUGP) show that these are found in the glycosome-containing small granular (SG) fraction (Fig 6A; Table 1).

Some of the IFM images (Figs 2, 3, and 4) show imperfect colocalization with the classical glycosomal marker in that the ratios of green NS enzyme- to red TbGAPDH-signals are variable. For example, some of the NS enzymes visually appear to occupy a subset of the TbGAPDH positive glycosomes (eg. for TbGNA, TbPMM, TbMPGT, TbUGP and TbGMER) whereas others show (green) glycosomes that contain less TbGAPDH (eg. for TbUAP, TbPMI, TbGALE). Similar results have been noted for bsf cells for TbPMM and TbPMI [42,46]. Nevertheless, quantitative analysis of the red and green channel signals across several cells (Table 3) shows that the majority of NS enzyme (green) signals colocalize with TbGAPDH (red) signals. Collectively, these data suggest some heterogeneity in glycosomal content, whereby the ratios of NS enzymes to TbGAPDH in individual glycosomes can vary quite widely. Whether this represents a mixture of immature and mature glycosomes or a mixture of different mature glycosomes, or a combination of both, is unclear from these data. However, glycosome biogenesis by growth and fission and by *de novo* biogenesis of nascent glycosomes that bud from the ER seems likely, as reviewed in [77]. Evidence for the latter can be found in our label-chase proteomic studies [54]. In this work, cells expressing GFP-tagged Pex13.1 were labelled to steady-

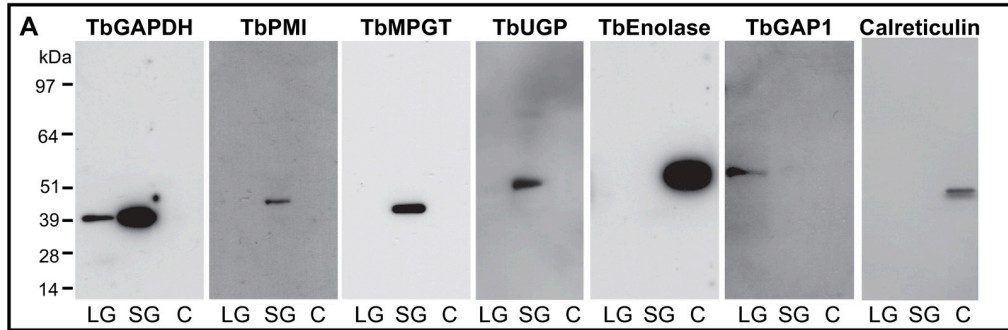
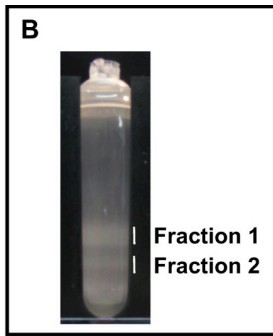

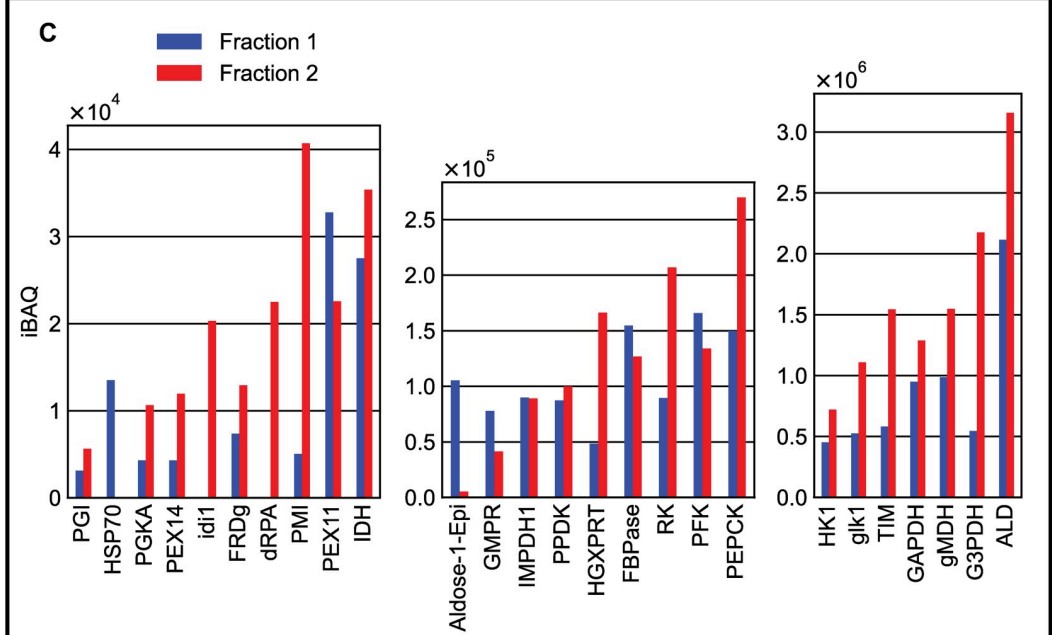

**Fig 6. Subcellular localization by subcellular fractionation and evidence for glycosome heterogeneity by density gradient centrifugation and proteomics.** Panel A: Pcf trypanosome lysate was submitted to differential centrifugation [70] and aliquots of the large granular (LG), small granular (SG) and cytosolic plus microsomal fractions (C) were subjected to SDS-PAGE and Western blotting with antibodies to authentic glycosomal (TbGAPDH), mitochondrial (TbGAP1), cytoplasmic (TbENO) and ER (calreticulin) markers and to NS biosynthetic enzymes (TbPMI, TbMPGT and TbUGP), as indicated. Panel B: The SG fraction was further fractionated by iodixanol density gradient centrifugation and two opalescent bands (fractions 1 and 2) were isolated. Panel 3: Aliquots of fractions 1 and 2 from panel B were subjected to proteomic analysis by LC-MS/MS. The relative abundances (iBAQ values over three scales, y-axes) are plotted against the identified proteins (TriTrypDB identifiers, x-axes).

state over 8 cell divisions with heavy isotope lysine and arginine and then placed into medium containing light lysine and arginine to perform a 5 h isotopic chase. Immediately after steady-state (heavy) labelling and after (light) chase, magnetic beads bearing anti-GFP antibodies were used to capture GFP-tagged Pex13.1 containing organelles which were taken for stable isotope in cell culture (SILAC) quantitative proteomics. At the steady-state label time point, Pex13.1 was found associated with several ER resident proteins as well as glycosome components. However, after the light isotopic chase period, the apparent association of Pex13.1 with ER proteins was greatly diminished, suggesting that Pex13.1 and ER resident proteins part company during the biogenesis of glycosomes. Consistent with this, Bauer *et al.* showed by IFM and sub-cellular fractionation that Pex13.1 accumulates in the ER in pcf cells grown in low glucose [78]. The same group has also demonstrated the induction of glycosome heterogeneity in response to extracellular glucose levels [79], suggesting that the *de novo* and growth

and division pathways of glycosome biogenesis are dynamic and responsive to changing environmental factors [77].

Also consistent with the aforementioned concept of glycosome heterogeneity, we found that we could resolve two glycosomal fractions by density gradient centrifugation (Fig 6B) and that the glycosomal protein contents of these two fractions were different (Fig 6C). In this analysis, TbPMI was identified in both fractions but was enriched in fraction 2, consistent with the presence of TbPMI-rich (green and yellow) and TbPMI-poor (red) glycosomes in the TbPMI immunolocalization images (Fig 2B). Interestingly, proteins likely to be involved in protein import (glycosomal HSP70 and Pex11) are enriched in fraction 1, suggesting immature and actively importing glycosomes may be enriched in this lower density (lower protein content) fraction.

Only one of the NS biosynthetic enzymes (TbPAGM) appears to have dual location, being found in the cytoplasm as well as the glycosomes by IFM and digitonin latency (Fig 2, Table 3 and Fig 5). Whether TbPAGM has some role in cytoplasmic metabolism or whether this PTS-less protein is simply imported inefficiently into the glycosome remains to be determined.

Our localisation of TbMPGT, which converts Man1P to GDP-Man, to the glycosomes is at variance with a previous report using bsf cells that suggested a cytoplasmic location [48]. However, in that case the antibody used was raised to *L. major* MPGT and it is also possible that different growth conditions or other factors might account for this discrepancy. Several IFM localisations reported here do not agree well with those reported in the high-throughput Tryp-Tag database [80]. In that case, eight epitope-tagged proteins, TbPGI, TbUGP, TbPAGM, TbUAP, TbPMI, TbPMM, TbGMD and TbGMER were variously described as flagellar cytoplasmic and/or nucleoplasmic and/or mitochondrial and/or cytoplasmic. However, C-terminal tagging of PTS1-containing TbPGI and TbPMI would likely cause their mislocalisation, and visual inspection of the IFM micrographs for N-terminally tagged TbUGP and TbPAGM suggests punctate staining reminiscent of glycosomes.

In summary, together with previously published sub-cellular fractionation data for TbHK1, TbPGI and TbGALE and glycosome proteome data for TbHK, TbPGI, TbUAP, TbGALE, TbUAP and TbPMI (Table 1), the data presented here provide a compelling case for the presence of the entire NS biosynthetic machinery in the glycosomes of both bsf and pcf *T. brucei*. This is radically different from other eukaryotes, where NS biosynthesis is either known or assumed to be located in the cytoplasm. The glycosomal location of NS biosynthesis in *T. brucei*, in turn, strengthens our postulate that an NS transporter (NST) or transporters is/are likely to exist in the bsf and pcf glycosome membranes to transport these large and negatively charged metabolites from the lumen of the glycosome into the cytoplasm. Once in the cytoplasm, NSs can be imported into the Golgi apparatus and ER by canonical SLC35-family NSTs [81] to perform glycosylation reactions in glycoprotein and glycolipid biosynthesis. The putative glycosomal NSTs may be quite different from the canonical SLC35-family NSTs since the direction of transport of the glycosomal NSTs (organelle lumen to cytosol) is in the opposite direction to that of the SLC35-family NSTs (cytosol to organelle lumen). It is possible that glycosmal NST(s) may provide a selective therapeutic target(s) for human and animal African trypanosomiasis.

## Methods

### Cell culture

Procyclic form *T. brucei* 427 strain containing T7 RNA polymerase and Tet repressor protein genes, respectively under control of G418 and hygromycin (clone 29.13.6 cells, kindly provided by George Cross) were grown at 28°C without $CO_2$ in original SDM-79 medium [82] containing hygromycin B (Roche) at 50 μg/ mL, G418 (Invitrogen) at 15 μg/ mL, 15% (v/v)

heat inactivated fetal bovine serum (FBS), 2 g/L sodium bicarbonate, fresh 2 mM Glutamax I (Invitrogen) and 22.5 mg / ml haemin (added from a 0.05 M stock in NaOH), adjusted to pH 7.3. Bloodstream form *T. brucei* 427 strain containing T7 RNA polymerase and Tet repressor protein, under control of G418, known as single markers cells, kindly provided by George Cross, were grown at 37˚C with 5% $CO_2$ in HMI-9T medium [83] containing fresh 2mM Glutamax I (Invitrogen) and 2.5 μg/ mL G418 (Invitrogen).

## Production of recombinant TbMPGT

The TbMPGT ORF (gene Tb927.8.2050) was amplified from genomic DNA by PCR and cloned into pET28a, containing a 6-His N-terminal tag with a thrombin cleavage site, and overexpressed in BL21 (DE 3) Gold *E. coli* grown at 37˚C in LB medium supplemented with kanamycin (50 μg / ml) and tetracycline (10 μg / ml). When the $OD_{600}$ reached about 2.0, the temperature was lowered to 16˚C and protein expression was induced with 1 mM IPTG (Formedium) for 22 h. Cells were harvested, resuspended in lysis buffer (20 mM Tris, 500 mM NaCl, 20 mM imidazole, pH 8.0), supplemented with DNAse I and EDTA-free protease inhibitors (Roche), disrupted using a French press and centrifuged at 30,000 g for 30 min at 4˚. The supernatant was filtered (0.2 μm) and applied to a 5 ml HisTrap HP chelating column (GE healthcare), pre-equilibrated in lysis buffer, and eluted with a linear imidazole gradient from 96% buffer A (20 mM Tris, 500 mM NaCl, pH 8) to 100% buffer B (20 mM Tris, 500 mM NaCl, 500 mM imidazole, pH 8). Fractions containing TbMPGT (which eluted around 300 mM imidazole) were pooled, concentrated, digested with thrombin (NEB) to remove the His-tag and subsequently purified on a Superose 12 10/300 column (GE healthcare) equilibrated with 10 mM Hepes, 100mM NaCl, pH 7.4. The yield, estimated by BCA assay, was about 10 mg of soluble, enzymatically active and highly purified TbMPGT per litre of culture (S1 Fig). The identity of the recombinant TbMPGT was confirmed by proteomics.

## Production of mouse antibodies against recombinant nucleotide sugar biosynthetic enzymes

Aliquots of recombinant purified TbMPGT (S1 Fig), TbGALE [41], TbGNA [49], TbUAP [50] and TbGMER [52] were submitted to David's Biotech (Germany) for the immunization of two mice per protein, according to their recommended protocol using Freund's complete adjuvant for the primary immunisation with 0.1 mg protein per mouse and two secondary boosts with 0.05 mg per mouse in Freund's incomplete adjuvant. Pooled mouse anti-TbMPGT, anti-TbUAP and anti-TbGNA sera were immunopurified by affinity chromatography on Sepharose4B beads coupled to recombinant TbMPGT, TbUAP and TbGNA, respectively, at 3–5 mg / ml gel. The affinity chromatography beads were prepared using CNBr-Sepharose 4B (GE Healthcare) according to the manufacturer's instructions. The coupled beads were stored in phosphate buffered saline (PBS) containing 0.05% (w/v) sodium azide. For affinity purification, pooled mouse sera were centrifuged at 16,000 g for 15 min at 4˚C and the supernatants mixed with an equal volume of PBS. These were incubated with the corresponding affinity chromatography beads for 2 h with rotation at 4˚C. The beads were washed with PBS until the optical density at 280 nm (OD280) of the supernatant was below 0.05 and the specifically bound antibodies were then eluted with 50 mM sodium citrate pH 2.8. The eluates were immediately neutralized with 1M Tris-HCl pH 8.8 and quantitated by OD280.

## Immunoprecipitation

*T. brucei* bsf cells were washed three times with trypanosome dilution buffer (TBD; 20 mM $Na_2HPO_4$, 2 mM $NaH_2PO_4$, 80 mM NaCl, 5 mM KCl, 1 mM $MgSO_4$, 20 mM glucose pH 7.7)

and lysed at $1 \times 10^9$ cells/ml in 1% (w/v) SDS in 20 mM Tris-HCl pH 6.8 containing 0.1 M DTT and heated at 50°C for 15 min. The SDS lysate was diluted to 0.03% SDS with 1% (w/v) Triton X-100 in 20 mM Tris-HCl pH 6.8, 0.15 M NaCl, 0.1 mM TLCK, 1 μg / ml leupeptin, 1 μg / ml aprotinin and 1 mM PMSF. Insoluble material was removed by centrifugation at 16,000 g for 10 min at 4°C and an aliquot of the supernatant (equivalent to $2 \times 10^8$ cells) was incubated with 5 μL of rabbit anti-TbGalE antiserum [44] for 1 h at 4°C and immunoprecipitated with 50μL of Protein A magnetic beads (Invitrogen). A similar procedure was used for TbGMER immunoprecipitation, but instead mouse antisera anti-GMER and protein G magnetic beads (Invitrogen) were used. Samples were run in SDS-PAGE for Western blotting.

## Western blotting

Samples were run on 4–12% gradient NuPAGE gels (Invitrogen) and Western blotted onto nitrocellulose using an iBlot system (Invitrogen). The pieces of nitrocellulose were either stained with Ponceau red for protein or blocked with 30 ml pre-filtered (20 μm) blocking buffer (50 mM Tris-HCl pH 7.4 containing 0.15 M NaCl, 0.25% BSA (w/v), 2% (w/v) fish skin gelatin and 0.05% (w/v) Tween 20) using a SNAPid system (Millipore). The latter were incubated with 0.5–1 μg / ml affinity purified antibody or with 1:1,000 diluted mouse antiserum (both diluted in blocking buffer) against the respective nucleotide sugar biosynthetic enzymes for 1 h at room temperature. Using the SNAPid system, the blots were washed 3 times 30 ml PBS, 0.1% Tween 20 and then incubated with 5 ml anti-mouse HRP (Invitrogen) diluted 1:10,000 in blocking buffer without sodium azide. The blots were subsequently washed with PBS, 0.1% Tween 20, removed from the SNAPid system and developed with ECL (Thermo-Fisher). Rabbit anti-TbGAPDH, rabbit anti-TbEnolase, rabbit anti-TbPEX13 and rabbit anti-TbAldolase (kind gifts from Prof. Paul Michels, Univ. of Edinburgh, UK) were used at 1:5,000, 1:5,000, 1:10,000 and 1:5,000 dilutions in blocking buffer, respectively. In addition, developments with mitochondrial marker [rabbit anti-TbGAP1 (a kind gift from Prof. Julius Luke, Institute of Parasitology, Czech Republic)] were performed at a dilution of 1:1,500 and developments with ER marker [immunopurified rabbit anti-calreticulin (Stressgen, Enzo LifeSci, USA)] were performed at a dilution of 1:1,000. Blots were developed with 1:10,000 anti-rabbit HRP (Invitrogen) and ECL reagents according to the manufacturer's instructions (Thermo-Fisher). The blots were exposed onto ECL film (GE Healthcare Amersham), developed and scanned in an Epson office scanner.

## Immunofluorescence microscopy

Trypanosomes were washed in ice-cold phosphate buffer saline (PBS) for pcf cells or ice-cold trypanosome dilution buffer (TDB; 20 mM $Na_2HPO_4$, 2 mM $NaH_2PO_4$, 80 mM NaCl, 5 mM KCl, 1 mM $MgSO_4$, 20 mM glucose, pH 7.7) for bsf cells and resuspended in the same buffer. Equal volumes of 8% paraformaldehyde (PFA) in PBS or TDB, respectively, were added and cells were fixed at 4° C. Subsequently, suspensions of fixed cells were spotted on cover slips, allowed to air dry, permeablized with 0.1% TX100 in PBS for 10 min and blocked in PBS, 5% fish skin gelatin, 10% normal goat serum and 0.05% TX100. Affinity purified anti-TbMPGT, anti-TbUGP and anti-TbGNA mouse antibodies were used at 0.5–1.0 μg / ml diluted in PBS, 1% fish skin gelatin, 0.05% TX100 and incubated for 1 h at room temperature in a humid chamber. Anti-TbGALE, anti-TbPAGM, anti-TbPMI, anti-TbPMM and anti-TbGMER mouse sera were diluted at 1:500 or 1:1,000 in the same buffer. Rabbit anti-GAPDH and rabbit anti-TbEnolase sera (generous gifts of Dr Paul Michels) were used at 1:2,000 and 1:4,000 dilution, respectively. Coverslips were washed in PBS, 1% fish skin gelatin, 0.05% TX100 and incubated for 1 h at room temperature with 50 μL goat anti-mouse IgG Alexa 488 conjugated

mixed goat anti-rabbit IgG Alexa 594 conjugated, both diluted at 1:500 in same buffer. Coverslips were washed and mounted on glass slides using Prolong Gold (Invitrogen). Microscopic images were obtained in a Zeiss LSM 710 META confocal microscope.

Colocalisations were quantified using Volocity software (Quorum Technologies). Zeiss.lsm images were imported into the Volocity library and processed as follows: The thresholds for the red and green channels were calculated from a region of interest outside the cell images. Individual cells were selected using the region of interest (ROI) tool and colocalization statistics were calculated and recorded for each cell. Mander's colocalisation coefficients (the fraction of green signal colocalising with red compartments) are reported in (Table 3), according to the recommendations of [84].

## Digitonin latency

Digitonin latency was performed as previously described in [42]. Briefly *T.brucei* bloodstream or procyclic form cells (2 x $10^9$ cells) were washed twice with 10 ml 250 mM sucrose, 25 mM Tris-HCl pH 7.4 and 1 mM EDTA (STE buffer) and resuspended in 1.5 ml STEN (STE containing 0.15 M NaCl). Aliquots (0.15 ml) were treated with equal volume of digitonin at various concentrations in the presence of 0.1 mM TLCK, 1 μg / ml leupeptin, 1 μg / ml aprotinin and 1 mM PMSF. Digitonin stock was prepared at 10 mg / ml in DMSO and diluted in STEN to the required concentrations. Complete extraction was obtained in parallel by treating an equivalent number of cells with 0.1% Triton X-100 in STEN. The lysates were incubated for 5 min at room temperature and the insoluble materials removed by centrifugation at 16,000 g for 2 min. The pellets were discarded and samples of the supernatants were run on a reducing 4–12% gradient NuPage gel (Invitrogen) and transferred to nitrocellulose for Western blotting using antisera produced in mouse and at the dilutions described in the immunofluorescence section. Different amounts of same samples were loaded onto separate gels for each antibody Western blot, in order to keep within the linear range of film exposure. For *Tb*PMM, TbGALE and TbGMER 1x$10^7$ cell equivalents per lane, for TbPAGM, TbMPGT, TbUGP, TbUAP, TbPMI, 5 x $10^6$ cell equivalents per lane, for TbGAPDH, TbAldolase (Aldo), for TbEnolase (Eno) 1x$10^5$ cell equivalents per lane and for TbPEX13 5x$10^6$ cell equivalents per lane.

## Subcellular fractionation by differential centrifugation followed by density gradient centrifugation

The pcf cells used for differential and density gradient centrifugation came from a stable isotope in cell culture (SILAC) labelling experiment. However, the isotopic labelling is not relevant to the specific experiments and results presented here. The starting pcf cell preparation (a total of 5.6 x $10^9$ cells washed in STE buffer) was a 1: 1 mixture of (i) GFP-tagged PEX13.1 pcf mutants labelled to steady state over 8 cell divisions, and to a final cell density of 1.1.x $10^7$ cells / ml, in R6K6 SDM-79 medium containing heavy isotopes of L-Arginine (R6) and L-Lysine (K6), and (ii) double marker cells grown in normal SDM-79 medium (R0K0) and also harvested at 1.1 x $10^7$ cells / ml.

The mixed cells were lysed using a One-shot high-pressure cell disruptor (Constant Systems, UK) operated at 10,000 psi as described in [54]. The lysate was submitted to differential centrifugation as described in [70]: After spinning at 1,500 g to the remove nuclei and any remaining intact cells and cell ghosts, the supernatant was submitted to centrifugation at 5,000 g to yield a Large Granular fraction (LG) pellet, enriched in mitochondria. Subsequent centrifugation of the supernatant at 15,000 g pelleted the Small Granular fraction (SG), enriched in glycosomes, with concomitant generation of a supernatant (C) enriched in cytosol and ER, Golgi and plasma membrane microsomes.

The SG fraction was subsequently submitted to density gradient centrifugation using iodixanol (Optiprep, Axis Shield, UK) according to [53], with some modifications: A discontinuous step gradient of iodixanol was prepared by layering 20, 25, 30, 35, 40 and 45% (w/v) iodixanol in STE buffer inside an Optiseal polyallomer tube (Beckman) using a peristaltic pump to apply each layer to the bottom of the tube, starting with the least dense layer and displacing it with next dense layer and so on. The SG fraction resuspended in STE buffer was loaded carefully on top of the gradient and centrifuged at 45,000 rpm using VTi-50 vertical rotor in a Beckman ultracentrifuge for 1h at 4˚C. After ultracentrifugation, the tube was punctured with needle and syringe to collect 2 distinct opalescent bands, fraction 1 (top band) and fraction 2 (lower band). The refractive index of each fraction was measured using a refractometer (Abbe refractometer model NAR-1T) and plotted against a standard curve of 20, 25, 30, 35, 40 and 45% (w/v) iodixanol in STE. The density was calculated according to published tables of iodixanol percentage versus density (g/mL).

### Proteomic analyses

Samples of fraction 1 and fraction 2 were reduced with dithiothreitol and alkylated with iodoacetamide and subsequently digested with trypsin and processed using FASP II (filter-aided sample processing), as described previously in [54]. Samples were desalted using a custom made Porous R2 columns packed with 1.5 mg Porous R2 reverse phase beads (ABI). The columns were pre-conditioned by washing with 90% $CH_3CN$ containing 0.1% formic acid followed by equilibration in 0.1% formic acid in water. Samples were applied in 0.1% formic acid and the columns washed with 0.1% formic acid then subsequently eluted three times with 90% $CH_3CN$ containing 0.1% formic acid. The eluates were combined, dried in a Speedvac vacuum concentrator, resuspended in 35% $CH_3CN$ containing 0.1% formic acid and fractionated using strong cation exchange (SCX) chromatography: The combined eluates were applied into a custom made SCX column made by adding Poros 50 HS SCX beads (ABI) to a SCX Ziptip (Millipore). Bound peptides were eluted sequentially with 20, 100, 150, and 200 mM NaCl in 35% $CH_3CN$ containing 0.1% formic acid, followed by final elutions with 0.5% $NH_4OH$ in 50% isopropanol and 50% isopropanol (IPA sample). Each of these five sub-fractions were dried and re-suspended in 50 μL 1% formic acid and 15 μL aliquots were subjected to liquid chromatography on Ultimate 3000 RSLCnano system (Thermo Scientific) fitted with an Acclaim PepMap 100 (C18, 100 μM x 2 cm) trap cartridge. Peptides were separated on an Easy-Spray PepMap RSLC C18 column (75 μM x 50 cm) (Thermo Scientific) using a linear gradient from 2 to 40% buffer B (80% acetonitrile, 0.1% formic acid) in buffer A (0.1% formic acid) over 70 min. The HPLC system was coupled to a Orbitrap Velos Mass Spectrometer (Thermo Scientific) with a source voltage of 1.2 kV.

### Proteomic data analysis

LC-MS/MS data were analysed for protein identification using MaxQuant 1.6.14 [85,86] with the in-built Andromeda search engine [87]. The raw files were searched against the *T. brucei brucei* 927 annotated protein sequences from TriTrypDB release 46 [88] supplemented with the *T. brucei brucei* 427 VSG221 (Tb427.*BES40.22*) protein sequence. The mass tolerance was set to 4.5 ppm for precursor ions and trypsin was set as proteolytic enzyme with two missed cleavage allowed. Carbamidomethyl on cysteine was set as fixed modifications. SILAC labelling [5] of heavy arginine (Arg-6) and Lysine (Lys-6) were specified. Oxidation of methionine and Acetylation of Protein N-term were set as variable modifications. The false-discovery rate for protein and peptide level identifications was set at 1%, using a target-decoy based strategy. The minimum peptide length was set to seven amino acids and protein quantification was

performed on unique plus razor peptides [89]. "Reverse Hits", "Only identified by site" and "Potential contaminant" identifications were filtered out. Only protein groups with at least two unique peptide sequences and Andromeda protein score greater than 5 were selected for further analysis. The high confidence glycosomal resident proteins were extracted from Güther *et al.*, 2014 [54]. Gene abbreviations were retrieved from the TriTrypDB database. The iBAQ values [89,90] were extracted from the MaxQuant output and visualised as bar graphs. The analysis pipeline was implemented in python using the SciPy packages (https://www.scipy.org/ ) and Jupyter notebook (https://jupyter.org/).

The mass spectrometry proteomics data have been deposited to the ProteomeXchange Consortium via the PRIDE [91] partner repository with the dataset identifier PXD023124. Reviewer account details: **Username:** reviewer_pxd023124@ebi.ac.uk. **Password:** qqm7vZg3. The analysis pipeline is available in GitHub (https://github.com/mtinti/nucleotide_sugar) and it is archived in Zenodo (https://doi.org/10.5281/zenodo.4289929). The analysis pipeline is reproducible using the mybinder app with the link reported in the GitHub repository.

## Supporting information

**S1 Fig. Recombinant TbMPGT.** Analysis by SDS-PAGE and Coomassie blue staining of TbMPGT expressed in *E.coli*, purified by nickel chromatography, digested with thrombin to remove the N-terminal 6-His tag and then purified by FPLC gel-filtration. Molecular weight standards are shown on the left.
(DOCX)

**S2 Fig. Mono-specificity of antibodies to nucleotide sugar biosynthetic enzymes used in this study.** The mouse polyclonal antibodies raised against TbMPGT, TbGNA, TbUAP, TbGALE and TbGMER (Tables 1 and 2) were used in Western blotting with an anti-mouse-HRP secondary antibody. The lanes contained either 5 x 10⁶ cell equivalents of *T. brucei* bsf total cell lysates (lanes 1, 2, 3, 5, and 7) or, when no signal was recorded against whole lysate due to low abundance, the lanes contained an immunoprecipitate from 2 x 10⁸ cell equivalents of total bsf lysate using rabbit anti-TbGALE [44] (lane 4) or mouse anti-TbGMER (lane 6). In each case, a single band* with an apparent molecular weight consistent with the target antigen was recorded, demonstrating the mono-specificity of the antibodies. *Note: the two additional bands marked by asterixis in (lane 6) are due to mouse IgG heavy and light chains from the immunoprecipitation. A Ponceau red stain of a Western blot lane containing 5 x 10⁶ cell equivalents of *T. brucei* bsf total cell lysate is shown in (lane 8). The positions of MW standards are shown on the right.
(DOCX)

## Acknowledgments

We thank Paul Michels, University of Edinburgh, for helpful discussions and for supplying rabbit anti-TbGAPDH, rabbit anti-TbEnolase, rabbit anti-TbAldolase, rabbit anti-Pex13. We also thank former members of MAJF lab: Karina Marino for supplying TbGNA, TbUGP and TbPAGM proteins and Matt Stokes for TbUAP in order to raise antibodies. We thank Fred Opperdoes for pointing out that TbGALE has a PTS1 signal many years ago.

## Author Contributions

**Conceptualization:** Maria Lucia Sampaio Guther, Michael A. J. Ferguson.

**Data curation:** Maria Lucia Sampaio Guther, Alan R. Prescott, Michele Tinti.

**Formal analysis:** Maria Lucia Sampaio Guther, Alan R. Prescott, Michele Tinti, Michael A. J. Ferguson.

**Funding acquisition:** Michael A. J. Ferguson.

**Investigation:** Maria Lucia Sampaio Guther, Alan R. Prescott.

**Methodology:** Maria Lucia Sampaio Guther, Alan R. Prescott, Sabine Kuettel.

**Project administration:** Maria Lucia Sampaio Guther, Michael A. J. Ferguson.

**Resources:** Maria Lucia Sampaio Guther, Alan R. Prescott, Sabine Kuettel, Michele Tinti.

**Software:** Alan R. Prescott, Michele Tinti.

**Supervision:** Maria Lucia Sampaio Guther, Michael A. J. Ferguson.

**Validation:** Maria Lucia Sampaio Guther, Alan R. Prescott, Michele Tinti.

**Visualization:** Maria Lucia Sampaio Guther, Alan R. Prescott, Michele Tinti.

**Writing – original draft:** Maria Lucia Sampaio Guther, Alan R. Prescott, Michele Tinti, Michael A. J. Ferguson.

**Writing – review & editing:** Maria Lucia Sampaio Guther, Michael A. J. Ferguson.

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
