## [Decision Letter · Decision Letter 0]

10 Jul 2020

Dear Prof. Ferguson,

Thank you very much for submitting your manuscript "Nucleotide sugar biosynthesis occurs in the glycosomes of procyclic and bloodstream form Trypanosoma brucei." for consideration at PLOS Neglected Tropical Diseases. As with all papers reviewed by the journal, your manuscript was reviewed by members of the editorial board and by several independent reviewers. In light of the reviews (below this email), we would like to invite the resubmission of a significantly-revised version that takes into account the reviewers' comments. 

We cannot make any decision about publication until we have seen the revised manuscript and your response to the reviewers' comments. Your revised manuscript is also likely to be sent to reviewers for further evaluation.

Sincerely,

Armando Jardim, PhD

Associate Editor

S Madison-Antenucci

Deputy Editor

Reviewer's Responses to Questions

**Key Review Criteria Required for Acceptance?**

**Methods**

-Are the objectives of the study clearly articulated with a clear testable hypothesis stated?

-Is the study design appropriate to address the stated objectives?

-Is the population clearly described and appropriate for the hypothesis being tested?

-Is the sample size sufficient to ensure adequate power to address the hypothesis being tested?

-Were correct statistical analysis used to support conclusions?

-Are there concerns about ethical or regulatory requirements being met?

Reviewer #1: In this paper, the authors demonstrate that enzymes in the nucleotide sugar biosynthetic pathway are localized ot glycosomes. Methods include immunofluorescence assays (IFA) and digitonin fractionations. Sufficient information is provided for the analysis of IFA. I presume that the gels in figure 3 are representative of three biological replicates but I could not find this explicitly stated. Can the authors include the number of replicates in either the Materials and methods or the figure legends? In lines 710-712 the authors state that a portion of the figure has been reproduced from another manuscript. I've not seen this done before and am unsure how to evaluate this data. Perhaps the authors could state why they are using previously published work instead of data from their own experiments. Also, it would be helpful if the authors identified these data visually in the figure as I wasn't able to discern exactly which parts were previously published. Is this portion of the figure is necessary to show that digitonin is solubilizing cell membrane but that glycosome components are still protected? If so, the enolase and aldolase panels are essential for those controls. Can the authors clarify this issue? In lines 164-166 the authors state that aldolase and GAPDH are not quantitatively released until addition of Triton because of the "aggregated nature of these very abundant proteins in the dense crystalloid glycosome core" While many have speculated that the dense nature of the glycosomes is responsible for some of the unique behaviors of glycosome proteins, this hasn't been experimentally demonstrated. For this reason, it may be desirable to qualify the statement to "it has been proposed that ....". I agree that this "non-quantitive" release is common but I don't know that the cause of this is understood. I am always uncomfortable when proteins are not detectable at all in digitonin experiments e.g. TbGMER, GAPDH, PMM. While not likely, it is possible that the proteins are cytosolic but digitonin-sensitive. Appearance of the proteins in the digitonin-insoluble fraction would be comforting. However, I understand that this is not commonplace for these assays and it is not fair for me to ask the authors to include that experiment. Because the authors provide two independent methods to demonstrate localization (IFA and digitonin fractionations), I am comfortable with the conclusion that the proteins are glycosomal.

Reviewer #2: Methods are acceptable.

Reviewer #3: See comments below

**Results**

-Does the analysis presented match the analysis plan?

-Are the results clearly and completely presented?

-Are the figures (Tables, Images) of sufficient quality for clarity?

Reviewer #1: The data does match the analysis plan. With the exception of issues described above in which previously published data was included, the figures are clear and suitable for publication.

Reviewer #2: See my overall review in the Editorial and Data Presentation Modifications section.

Reviewer #3: See comments below

**Conclusions**

-Are the conclusions supported by the data presented?

-Are the limitations of analysis clearly described?

-Do the authors discuss how these data can be helpful to advance our understanding of the topic under study?

-Is public health relevance addressed?

Reviewer #1: It is my opinion that the conclusions are supported by the data, that the authors discuss how these data can be helpful to advance our understanding of parasite metabolism, and that public health relevance is addressed (see summary and general comments). The authors provide a compelling explanation for conflicting data in regards to the localization of several of the proteins that are included in the Tryptag database. The explanation regarding the previous finding of TbMPGT in the cytoplasm is less compelling as the authors suggest the discrepancy is due to using antibodies raised against Leishmania MPGT (Denten et al. PMID:19919534). While the IFA signal in the Denten et al paper may have been a result of non-specific interactions (there is no control with MPGT-deficient parasites), the fractionations indicate that most of the signal was found in the cytosolic fraction with a small amount found in microsomes. While there does appear to be a difference in size between the cytosolic and microsomal species the westerns yield a single, well-defined band. Is there another explanation? Is it possible that this protein could exhibit dual localization under different growth conditions or in different strains where perhaps import of the protein is inefficient?

Reviewer #2: See my overall review in the Editorial and Data Presentation Modifications section.

Reviewer #3: See comments below

**Editorial and Data Presentation Modifications?**

Reviewer #1: Minor editorial suggestions: Qualify the statement in lines 163-166 to highlight that people have hypothesize that the non-quantitative release often observed with glycosome proteins is due to the aggregated nature of these proteins. 

--Ln 286 should read "His-tag"?

--Lines 308,310: OD280 formatting

--Line 314-TBD buffer composition is not given

--Line 682 T. brucei should be italicized

--Lines 710-712 and Figure 3: clarify which regions of the figure were reproduced from previous work

--Figure 3--red box around text

Reviewer #2: Many proteins are modified by the addition of different sugars. Sugar addition is accomplished by different enzymes that use different activated nucleotide sugars to distinctively modify proteins. In most eukaryotic cells, these enzymes are localized to the cytosol; however, in the trypanosomatids, including the trypanosome Trypanosoma brucei that causes African sleeping sickness, some of these enzymes have been localized to the glycosome, a specialized member of the peroxisome family of membrane-enclosed organelles. In this manuscript, Güther and colleagues perform a comprehensive analysis of the subcellular localization of all the sugar transferases in T. brucei in both its insect (procyclic) and bloodstream forms. Using primarily immunofluorescence microscopy and latency of protein release following treatment of cells with different amounts of the detergent digitonin, the authors provide evidence that the sugar transferase enzymes of T. brucei are localized preferentially to glycosomes. However, this evidence is preliminary and essentially descriptive, and would be greatly strengthened by additional data and analysis.

Major points:

1) The images in Figure 2 appear to be maximum intensity projections. Planar slices should be shown. Moreover, quantification should be performed and the quantification data presented to show the amount of colocalization between a particular enzyme and the glycosomal marker GAPDH.

2) Subcellular fractionation must be done to show that a particular enzyme cofractionates with a known glycosomal marker and not with markers of other organelles such as the mitochondrion or ER.

3) Fig. 3A. As stated on lines 710-712, images in Figure 3A are taken from a previous publication. This is not acceptable. The authors should repeat the experiments.

Minor points:

1) Add scale bars to Figure 2.

2) Figure S1 can be removed.

3) Line 76. “... data suggest...” NOT “...data suggests...”.

4) Lines 158 and 159. There is some controversy that TbPEX13.1 is transmembrane. Best to rephrase simply as ‘membrane’.

5) Line 175. The word ‘notion’ is inappropriate. Replace with ‘evidence’ and remove the ‘The’.

6) Line 215. ‘pulse-chase’ NOT ‘label-chase’.

7) Line 217. ‘chase away’ is not scientific. Rephrase.

8) Line 219. “The same group has...” NOT “the same group have...”.

9) Line 229. “...which converts...” NOT “...that converts...”.

10) Line 681. ‘mutase’ NOT ‘mutate’.

11) Line 698. “...forms of T. brucei.” NOT “...form T. brucei.”

Reviewer #3: See comments below

**Summary and General Comments**

Reviewer #1: My interpretation of the authors conclusions include two significant findings. First, this work adds to the growing literature suggesting that glycosomes are heterogeneous as many of the glycosome proteins exhibit distinct localization patterns and do not exhibit complete colocalization. While the appreciation of glycosome variation is growing, findings such as these add to the list of proteins that may exhibit distinct localization patterns and are essential to understanding the basis of such heterogeneity. Second, the authors conclude that the NS pathway resides in glycosomes. Such localization necessitates a membrane transporter to facilitate the transfer of these metabolites to the cytoplasm and such a transporter would be an excellent drug target. The next step, identification of the NS transporter, is essential and challenging as there are no clear homologs in the genome and proteomics of the glycosome membrane has revealed few candidates. However, the evidence here that the NS pathway is glycosomal provides additional findings to support premise that such as transporter exists and that the search to identify it is worthwhile.

Reviewer #2: See my overall review in the Editorial and Data Presentation Modifications section.

Reviewer #3: In this original research article, Güther et al., building on a substantial body of previous work, have explored the subcellular location of nucleotide sugar biosynthetic enzymes in trypanosomes. Using IFA against procyclic form trypanosomes (previous worked explored the localization of these proteins by IFA in bloodstream form parasites) they have shown that the majority of these NS biosynthetic enzymes have a punctate distribution that colocalizes, in part with the glycosome marker protein GAPDH. In a separate experiment using differential solubilization of cellular membranes by digitonin, the location of these proteins to an intracellular subcompartment was confirmed. Thus, it is highly likely based upon these studies, as well as the substantial body of previously published work that includes glycosome proteomics, bioinformatic analyses, IFA, and digitonin latency experiments, that the NS biosynthetic enzymes colocalize to the glycosome. While the observations in this manuscript are not particularly novel, since 9 of the 13 NS biosynthetic enzymes had previously been observed to have a glycosome location, this paper does supply important confirmatory evidence for the unusual compartmentalization of these enzymes, which makes them an outlier in comparison to other eukarya.

The manuscript is fairly limited in terms of the types of the types of experiments and the results produced, but does contain a significant body of experimental work when one considers that the authors produced 5 de novo antibodies, and conducted IFA and digitonin latency experiments on 9 different antisera. Most of the experiments were executed with the appropriate controls (see comment below) and the data generated were relatively straight-forward in their interpretation, though there are a few concerns and minor comments for the authors to address below.

1) Since the bulk of the evidence in this manuscript relies on the immunofluorescence localization studies, the inclusion of more representative images is a must. The evaluation of subcellular distribution and colocalization with GAPDH based on just one image per NS biosynthetic enzyme is inadequate. Moreover, given the considerable amount of non-overlap between GAPDH and the nucleotide sugar enzymes, this seems particularly important. 

2) Along those lines, it appears from the images that some of the NS biosynthetic enzymes show a minor distribution adjacent to or even along the flagellum, and this appears distinct from GAPDH. Can the authors comment on whether this is the case or perhaps an artifact of the images shown here? 

3) I agree that the data certainly indicate glycosome colocalization for the NS biosynthetic enzymes, however, the distinct distribution from GAPDH is intriguing. While the authors indicate this is likely due to glycosome heterogeneity, I am curious whether the authors have looked at other markers? Either glycosomal or perhaps other subcellular compartments?

4) For the digitonin latency studies with BSF and PCF parasites described in Figure 3 A and B, the distribution of TbPAGM seems more distinctly glycosomal in PCF parasites. Also, I may be interpreting the figure legend incorrectly, but it suggests that the bottom two control panels for the BSF digitonin experiments, aldolase and enolase, were included from previous work? I don't consider data generated in a previous experiment to be an adequate control for use in this experiment.

5) Typographical error ln 300, TbUGP should read TbUAP.

PLOS authors have the option to publish the peer review history of their article (what does this mean?). If published, this will include your full peer review and any attached files.

Reviewer #1: Yes: Meredith Teilhet Morris

Reviewer #2: No

Reviewer #3: No
---

## [Editor Report · Decision Letter 1]

9 Nov 2020

Dear Prof. Ferguson,

Thank you very much for submitting your manuscript "Nucleotide sugar biosynthesis occurs in the glycosomes of procyclic and bloodstream form Trypanosoma brucei." for consideration at PLOS Neglected Tropical Diseases. As with all papers reviewed by the journal, your manuscript was reviewed by members of the editorial board and by several independent reviewers. In light of the reviews (below this email), we would like to invite the resubmission of a significantly-revised version that takes into account the reviewers' comments. 

We cannot make any decision about publication until we have seen the revised manuscript and your response to the reviewers' comments. Your revised manuscript is also likely to be sent to reviewers for further evaluation.

Sincerely,

Armando Jardim, PhD

Associate Editor

S Madison-Antenucci

Deputy Editor

Dear Professor Ferguson,

We thank you for the revisions addressing many of the reviewers issues. However, at this time it will not be possible to accept the manuscript in its current state since a critical point raised by by two reviewers with extensive cell biology and organelle biogenesis were not addressed. This is an important issue since the major thrust of the manuscript is to assign the localization of NS biosynthesis enzymes. Given the fact that the localization of some NS biosynthesis enzymes have been previously reported to have an incorrect localization based on IFM studies ist is critical to validate the findings that you report in this manuscript using subcellular localization as recommended by Reviewer #2 and that additional marker enzymes for ER and mitochondria should be included. The need for these additional experiments is further emphasized by the issue raised by Reviewer # 3, "Along those lines, it appears from the images that some of the NS biosynthetic enzymes

show a minor distribution adjacent to or even along the flagellum, and this appears distinct from GAPDH. Can the authors comment on whether this is the case or perhaps an artifact of the images shown here?" The rebuttal to this comment 

">>>We think this is an artefact of the wide range of NS enzyme : TbGAPDH signal ratios (evident from Table 3) and this is now discussed." is a little concerning as does not sufficiently address the reviewer's concerns. It is acknowledge that IFM and digitonin latency is frequently used (right or wrong), a more robust approach would be to include a subcellular fractionation experiment as indicate by the Reviewers. A careful examination of the Reviewer comment indicate that a subcellular fractionation experiment was a MUST.
---

## [Editor Report · Decision Letter 2]

12 Jan 2021

Dear Prof. Ferguson,

We are pleased to inform you that your manuscript 'Nucleotide sugar biosynthesis occurs in the glycosomes of procyclic and bloodstream form Trypanosoma brucei.' has been provisionally accepted for publication in PLOS Neglected Tropical Diseases.

Best regards,

Armando Jardim, PhD

Associate Editor

S Madison-Antenucci

Deputy Editor

---

## [Editor Report · Acceptance letter]

10 Feb 2021

Dear Prof. Ferguson,

We are delighted to inform you that your manuscript, "Nucleotide sugar biosynthesis occurs in the glycosomes of procyclic and bloodstream form Trypanosoma brucei.," has been formally accepted for publication in PLOS Neglected Tropical Diseases.

Best regards,

Shaden Kamhawi

co-Editor-in-Chief

Paul Brindley

co-Editor-in-Chief
